# Enantioseparation and Determination of Penconazole in Rat Plasma by Chiral LC-MS/MS: Application to a Stereoselective Toxicokinetic Study

**DOI:** 10.3390/molecules25132964

**Published:** 2020-06-28

**Authors:** Siman Ma, Jia Lun, Yanru Liu, Zhen Jiang, Xingjie Guo

**Affiliations:** Lab of analytical chemistry, School of Pharmacy, Shenyang Pharmaceutical University, No. 103 Wenhua Road, Shenhe District, Shenyang 110016, China; Mandy13188204161@163.com (S.M.); lunjia9665@163.com (J.L.); lyr2674990839@163.com (Y.L.)

**Keywords:** penconazole enantiomers, liquid chromatography-tandem mass spectrometry, enantioselective toxicokinetics

## Abstract

In this study, a specific and sensitive method of liquid chromatography coupled with tandem mass spectrometry (LC-MS/MS) was developed for the determination of penconazole enantiomers in rat plasma. The enantioseparation was achieved on a Chiralpak IC column by using acetonitrile/water (80:20, *v*/*v*) as the mobile phase. Penconazole enantiomers and internal standard l-lansoprazole (IS) were detected in multiple reaction monitoring (MRM) mode with positive electrospray ionization source. The method was validated over the concentration range of 2.5–250.0 ng mL^−1^ for penconazole enantiomers. Good linearity was obtained for both enantiomers with correlation coefficients (*R*) greater than 0.995. The relative error was well within the admissible range of −1.1–3.2%, and relative standard deviation was less than 6.0%. After validation, the established method was successfully applied to a stereoselective toxicokinetic study in female and male rats after oral administration of 50 mg kg^−1^ racemic penconazole. This is the first experiment regarding the stereospecific toxicokinetic study of penconazole and the bioanalytical approach for its quantitation in vivo.

## 1. Introduction

With the development of agricultural industrialization, the widespread use of pesticides plays an important role in increasing agricultural production in today’s world. However, the excessive use of pesticides can also cause serious pesticide residue problems in agricultural products, which has gradually become a hot spot of social concern [1,2,3]. Nowadays, people are exposed daily to low-level pesticides predominantly through dietary intake of contaminated food. Long-term pesticide accumulation in living organisms can cause potential harm to the human body. According to the research by Trösken et al., after exposure to azole fungicides, many azoles inhibited not only CYP51, but also CYP19, which is an important aromatase in estrogen biosynthesis [4]. For these reasons, toxicological risk assessment of pesticides is greatly needed [5]. Simultaneously, as an indispensable part of toxicological risk assessment, toxicokinetic study is also necessary for describing absorption, distribution, metabolism, and elimination (ADME) of some drugs that are potentially toxic to humans in vivo [6]. A large number of pesticides are chiral compounds with one or more chiral centers, and the enantiomers of chiral pesticides may differ in biological activities and toxicities. Han et al. [7] and Cui et al. [8] found that the fungicidal activities of (−)-hexaconazole and R-(−)-tebuconazole were significantly higher than their corresponding (+)-and S-(+)-enantiomers. Zhang et al. [9] revealed that comparing with (S)-flutriafol, (R)-flutriafol was more markedly toxic. Therefore, the enantiomeric behaviors of chiral pesticides should be investigated separately.

Penconazole (Figure 1) is a chiral triazole fungicide widely used as a racemate in agriculture to control and regulate powdery mildew in the growing period of fruits and vegetables [10]. Although the relevant research has confirmed that the fungicide is less toxic to mammals, the accumulation of penconazole for a long time in living organisms could also bring potential side effects in humans [11] and animals [12,13], such as cardiac oxidative damage, and structural and functional testicular impairment in rats. According to the study by Meng et al., rac-penconazole and its enantiomers showed different toxicological effects on liver function in mice [14]. Biochemical and histopathological analyses showed that exposure to rac-penconazole and (−)-penconazole led to significant liver damage and inflammation. However, exposure to (+)-penconazole treatment did not cause adverse effects on liver function and inflammation. In addition, Wang et al. [15] investigated the enantioselective dissipation of penconazole in cucumber, tomato, head cabbage, and pakchoi by field experiments. The results demonstrated that enantioselective dissipation occurred in head cabbage and pakchoi, with the preferential degradation of (−)-penconazole, resulting in an enrichment of the (+)-penconazole residues. Recently, Zhang et al. [10] studied the terminal residues of penconazole enantiomers in grape and soil and (−)-penconazole was found to degrade faster than its (+)-enantiomer in two matrices. However, thus far, there has been no research regarding the stereospecific toxicokinetic study of penconazole and the bioanalytical approach for its quantitation in vivo.

Zhang et al. [10] reported the enantioseparation of rac-penconazole under normal-phase mode by using cellulose-based chiral stationary phase (CPS) Chiralpak IC. The study demonstrated that the Chiralpak IC column showed superior enantioselectivity toward rac-penconazole (α ≥ 1.46 and R ≥ 6.59), applying *n*-hexane or petroleum ether as mobile phase and ethanol or isopropanol as a modifier. Zhao et al. [16] separated penconazole enantiomers in reversed-phase mode using an amylose-based CSP (Chiralpak IG) with acetonitrile and ultrapure water containing ammonium acetate and formic acid as mobile phase, and the retention times of (+)-and (−)-penconazole were 16.8 min and 18.1 min, respectively. In addition, one group [15] studied the enantioseparation of rac-penconazole on the cellulose-based CSPs (Lux Cellulose-2). In the optimized condition, penconazole enantiomers achieved basic separation under reversed-phase conditions.

In this work, a novel and sensitive LC-MS/MS method was developed for the determination of penconazole enantiomers in rat plasma. In the optimized condition, considering the better compatibility of mobile phase composition with mass spectrometry, a reverse mode was used for the separation. Comparing with Chiralpak IG, Chiralpak IC provided not only more less analysis (within 13 min) time, but also higher resolution (Rs = 2.21). The Chiralpak IC column was selected to separate penconazole enantiomers in reversed-phase mode. The chromatographic conditions, including the kind and percentage of organic modifier, were optimized. The proposed method was validated in compliance with the USA Food and Drug Administration (FDA) guidelines [17]. Finally, the assay method was applied to the enantioselective toxicokinetic studies of penconazole enantiomers in rat plasma after oral administration of 50.0 mg kg^−1^ racemic penconazole to male and female rats. The toxicokinetic parameters of each enantiomer were analyzed and compared.

## 2. Experimental

### 2.1. Chemicals and Reagents

Racemic Penconazole (purity: 98.0%) was purchased from Shanghai Pesticide Research Institute Co. Ltd. (Shanghai, China). The enantiomerically pure standard of (+)-penconazole was supplied by Shanghai Chiralway Biotech Co. Ltd. (Shanghai, China). l-lansoprazole (Internal standard, IS, 99.0% purity) was obtained from Henan Zhongshuai Pharmaceutical Technology Co. Ltd. (Zhengzhou, Henan, China). HPLC-grade acetonitrile (ACN) was supplied by Sigma-Aldrich (St Louis, MO, USA). Ultrapure water was purified with a Milli-Q water system.

### 2.2. Preparation of Standard and Quality Control Solutions

The stock solution of rac-penconazole and l-lansoprazole (IS) were prepared independently in acetonitrile at the concentration of 1.0 mg mL^−1^. The stock solution of rac-penconazole was further diluted in acetonitrile to obtain a series of working standard solutions ranging from 25.0 to 2500.0 ng mL^−1^ for each enantiomer. The IS working solution with the concentration of 1000.0 ng·mL^−1^ was prepared by diluting the IS stock solution in acetonitrile. Calibration standards were obtained by spiking 20 µL of rac-penconazole working standard solutions into 200 μL of blank rat plasma. Then, the final plasma concentrations for each enantiomer were 2.5, 5.0, 10.0, 25.0, 50.0, 125.0, and 250.0 ng·mL^−1^. Quality control (QC) samples were prepared at 5.0, 50.0, and 200.0 ng·mL^−1^ for each penconazole enantiomer by the same steps. All the solutions were stored at 4 °C before use.

### 2.3. Sample Preparation

Each 200 µL plasma sample was transferred into a 1.5 mL polypropylene microcentrifuge tube, followed by the addition of 20 µL IS working solution and 600 µL acetonitrile. Then, the mixture was vortexed for 3 min to precipitate protein. After centrifugation at 13,000 rpm for 10 min, the supernatant was transferred into a clean 1.5 mL polypropylene microcentrifuge tube and evaporated to dryness under gentle nitrogen stream at room temperature. The residue was reconstituted with 200 μL of mobile phase, and centrifuged again for 10 min at 13,000 rpm. Finally, 10 μL of supernatant was injected into the analytical system for LC-MS/MS analysis.

### 2.4. Chiral LC-MS/MS Conditions

The chromatographic analysis was performed on an Acquity™ UPLC system (Waters Corp. Milford, MA, USA) equipped with an Acquity™ UPLC binary solvent management system, an Acquity™ UPLC autosampler and a thermostatic column compartment. The chiral separation of penconazole was achieved on Chiralpak IC column (250 mm × 4.6 mm, 5 μm) protected with a Chiralpak IC guard column (10 mm × 4 mm, 5 μm) (Daicel, Japan). The mobile phase was acetonitrile/water (80:20, *v*/*v*) with a flow rate of 0.6 mL min^−1^. Column temperature and autosampler temperature were set at 25 °C and 4 °C, respectively. The injection volume was 10 μL. The elution order of penconazole enantiomers was confirmed by directly injecting the (+)-penconazole into the HPLC-MS/MS system. The first eluted enantiomer was confirmed as (+)-penconazole, while the second one was (−)-penconazole in this study.

A triple quadrupole mass spectrometer (Waters, Milford, USA) coupled with an electrospray ionization (ESI) source was used for detection of penconazole enantiomers and l-lansoprazole in positive ionization mode. For two MRM transitions, the most sensitive transition was used for quantification, while the other transition was used for confirmation. The optimal MS parameters were set as follows: capillary voltage of 3.0 kV; source temperature of 500 °C; desolvation temperature of 150 °C; desolvation gas (nitrogen) flow of 1000 L·h^−1^; collision gas (argon) flow of 0.12 mL·min^−1^; nebulizer gas pressure of 7.0 bar. For penconazole, the cone voltage (CV) was set at 16 V and the ion pair transitions of *m/z* 284.05 > 69.74 and *m/z* 284.05 > 158.63 were used for quantification and identification, respectively. The corresponding collision energy (CE) was 14 eV and 24 eV, respectively. For IS, the cone voltage (CV) was set at 25 V and the ion pair transitions of *m/z* 371.02 > 252.82 and *m/z* 371.02 > 118.88 were used for quantification and identification, respectively. The corresponding CE was 8 eV and 12 eV, respectively.

### 2.5. Enantioselective Toxicokinetic Study

Twelve Sprague–Dawley rats used for experiment were purchased from Changsheng Biotechnology Co. Ltd. (Benxi, Liaoning, China) with a weight of 200–220 g. The animal study was undertaken in accordance with the ethics requirements and authorized by the Official Ethical Committee at Shenyang Pharmaceutical University (license no. SCXK-liao-20150001). The rats were kept in the environment with controllable air condition at a temperature of 25 °C, and natural light–dark cycle. The rats were divided into male and female groups with six animals each. All rats were fasted for 18 h, but free access to water before the experiment. Penconazole formulation was prepared with 0.5% sodium carboxymethyl cellulose. After oral administration of 50.0 mg·kg^−1^ rac-penconazole to rats, 300 μL of the orbital blood was collected into 1.5 mL of heparinized disposable polypropylene tube at predose (0 h), 0.083 h, 0.167 h, 0.25 h, 0.33 h, 0.5 h, 1 h, 2 h, 4 h, 6 h, 8 h, 12 h, 24 h. Obtained blood samples were centrifuged at 5000 rpm for 10 min, then the supernatant were collected and stored at −80 °C until analysis.

The plasma concentrations of the penconazole enantiomers at different times were calculated from the calibration curve. Toxicokinetic parameters, including the maximum plasma concentration (C_max_), the time to reach the maximum plasma concentration (T_max_), the half-life of elimination (t_1/2_), the area under the plasma concentration curve from 0 to 24 h (AUC_0–24_), the area under the plasma concentration curve from 0 to infinity (AUC_0–∞_), mean retention time from 0 to 24 h (MRT_0–24_), and apparent clearance (CLz/F), were obtained from DAS 2.0 software (Chinese Pharmacological Society) by means of the noncompartment model. All parameters were expressed as mean ± standard deviation (SD). The *t*-test (Prime 5 statistical software) was used to compare the toxicokinetic parameters of each group. *p* < 0.05 was considered to be statistically significant.

## 3. Results and Discussion

### 3.1. Optimization of the Chiral Separation

Based on the literature mentioned above [10,15,16], three polysaccharide-based columns (Chiralpak IA, Chiralpak IB and Chiralpak IC) were tested to separate penconazole enantiomers. In the case of Chiralpak IA, only partial separation (Rs = 0.52) of rac-penconazole was observed by using acetonitrile–water or methanol–water as the mobile phase. When using Chiralpak IB as the chiral column, there was no tendency to separate penconazole enantiomers. The results clearly showed that the Chiralpak IC column provided both better resolution (Rs = 2.21) and less analysis time (within 13 min) compared with the other two. Therefore, the Chiralpak IC column was selected in the next study. In addition, the effect of kinds and contents of organic modifiers on chiral separation was evaluated. As shown in Table 1, the t_R_ and Rs values of the penconazole enantiomers increased obviously with the decreasing of modifier (methanol or acetonitrile) content. Compared with methanol/water as eluant, the acetonitrile/water system not only achieved complete separation of penconazole enantiomers, but also required less analysis time. Moreover, acetonitrile as the organic modifier could obtain better peak shape and higher column efficiency. Accordingly, as a comprehensive consideration of resolution, retention time, and peak shape, the Chiralpak IC column with the mobile phase of acetonitrile–water (80:20, *v*/*v*) was chosen as the optimized separation conditions for penconazole enantiomers.

### 3.2. Optimization of Mass Spectrometry Conditions

The signal intensities of precursor and fragment ions were investigated in positive and negative ionization modes for development of a sensitive and stable mass spectrometry analytical method. The results presented that greater signal intensities and higher sensitivity for penconazole and IS were observed in the positive ionization mode with ESI source. In the Q1 full scan mode, protonated precursor [M + H]^+^ ions at *m/z* 284.05 and 371.02 were observed for penconazole and l-lansoprazole, respectively. The stable and satisfactory signal-to-noise ratio (S/N) of penconazole and IS were obtained at the fragment ions for *m/z* 69.74 and 252.82, respectively. Therefore, the predominate ion transitions at *m/z* 284.05 → 69.74 for penconazole enantiomers and *m/z* 371.02 → 252.82 for l-lansoprazole were selected in the following quantitative analysis. Other MRM parameters such as the cone voltage and collision energy were also selected according to the MS response of the analytes. The satisfactory MS responses of penconazole and IS were achieved when cone voltages were set at 16 V and 25 V, and collision energies were set at 14 eV and 8 eV, respectively.

### 3.3. Optimization of Sample Pretreatment Procedures

In order to obtain a stable and ideal extraction recovery, different sample pretreatment methods including solid-phase extraction (SPE), liquid–liquid extraction (LLE), and protein precipitation (PPT) were investigated in this study. For SPE, two kinds of cartridges, including a C18 cartridge (Agela, 100 mg/3 mL) and an OASIS HLB cartridge (Waters, 60 mg/3 mL), were studied. It was found that both cartridges gave low recovery for the extraction of target analytes (46% for C18 and 60% for HLB). In terms of the LLE method, several kinds of organic solvents, including *n*-hexane/isopropanol, diethyl ether/dichloromethane, and ethyl acetate, were investigated in the recovery experiment. In comparison with diethyl ether/dichloromethane, higher extraction recovery could be obtained when using *n*-hexane/isopropanol and ethyl acetate as the organic solvents (83% for *n*-hexane/isopropanol and 78% for ethyl acetate). However, a significant matrix effect occurred for the LLE method. Finally, the PPT method was tested for the pretreatment of the plasma samples. The present study evaluated the extraction efficiencies of four kinds of precipitation solvents, including acetonitrile, methanol, acetonitrile/methanol (1:1, v/v), and acetone (Appendix A). Results showed that the use of acetonitrile as the precipitation solvent gave higher extraction recovery for target analytes, and the corresponding matrix effect was found to be negligible. Therefore, acetonitrile was utilized as the optimum precipitation solvent in the following study. In addition, the volume of precipitation solvent (400–800 μL) and the vortex time (time for PPT, 1–5 min) were also optimized. Finally, the volume of acetonitrile was determined as 600 μL, and the mixture was vortexed for 3 min to precipitate protein. Under the above optimization of pretreatment procedure, the extraction recovery for penconazole and IS reached the maximum.

### 3.4. Method Validation

The method was validated according to the FDA’s Bioanalytical Method Validation [17], including analysis of the specificity and carry-over effect, linearity and the lower limit of quantification, accuracy and precision, extraction recovery, matrix effect, and stability.

#### 3.4.1. Specificity and Carry-Over Effect

The specificity of the proposed method was investigated by comparing the chromatograms of blank plasma from six different rats, blank plasma spiked with penconazole enantiomers at the lower limit of quantification (LLOQ, 2.5 ng mL^−1^ per enantiomer), and plasma samples obtained at 2 h after oral administration of 50 mg kg^−1^ rac-penconazole to female rats. The measurement of the carry-over was obtained by injecting the blank plasma samples after injecting the spiked plasma sample at the upper limit of quantification (ULOQ, 250 ng mL^−1^) level. The carry-over should be lower than 20% for the LLOQ peak area and 5% for the IS peak area. The typical chromatograms are shown in Figure 2. Seen from Figure 2, there were no endogenous interference peaks eluted at retention time of (+)-penconazole, (−)-penconazole, and IS in blank plasma, indicating the acceptable specificity of the method. Meanwhile, the carry-over of penconazole and IS met the standard requirements. This suggested that the carry-over effect was negligible in the LC-MS/MS system, and the ULOQ sample had no effect on the accurate determination of the next sample.

#### 3.4.2. Linearity and the Lower Limit of Quantification (LLOQ)

Eight calibration standards in duplicate at each concentration of penconazole enantiomers were analyzed on three consecutive days. Calibration curves were generated by plotting peak-area ratios of each enantiomer to IS (y) versus the nominal enantiomer concentration (ng mL^−1^, x) in the spiked samples by 1/x^2^ weighted least squares linear regression. Two calibration curves presented excellent linearity over the range of 2.5–250.0 ng mL^−1^ for (+)-and (−)-penconazole with the correlation coefficient (*R*) greater than 0.995. The typical calibration curves were utilized as follows: y = 0.04027x + 0.01749 (*R* = 0.9962) for (+)-penconazole and y = 0.03879x + 0.005201 (*R* = 0.9954) for (−)-penconazole.

The LLOQ was the minimum concentration of each enantiomer in rat sample with a signal-to-noise ratio of 5. Six replicate plasma samples spiked with 2.5 ng mL^−1^ for each enantiomer were determined in three separate validation batches for the analysis of the accuracy and precision at the LLOQ. The accuracy (RE) and precision (RSD) were 7.2% and 6.4% for (+)-penconazole, and 8.5% and 5.0% for (−)-penconazole, respectively. Thus, the LLOQ was 2.5 ng mL^−1^ for both enantiomers.

#### 3.4.3. Accuracy and Precision

The accuracy and precision of the method were evaluated by analyzing QC samples (5.0, 50.0, 200.0 ng mL^−1^ for each enantiomer) in six replicates within one day (intraday) and during three consecutive days (interday). Accuracy was expressed as the RE within 15% of the mean from the nominal concentration, and precision was expressed as the RSD within 15% at each QC level. The results of accuracy and precision are given in Table 2. It was found that the accuracy (RE) was well within the admissible range of −1.1%–2.7% for (+)-penconazole and −0.9%–3.2% for (−)-penconazole, whereas the precision (RSD) was below 6.0% and 3.9% for (+)-penconazole and (−)-penconazole, respectively. The results demonstrated that the precision and accuracy data fulfilled the desired criteria.

#### 3.4.4. Extraction Recovery and Matrix Effect

The extraction recoveries of the penconazole enantiomers and IS were evaluated at three QC concentration (5.0, 50.0, 200.0 ng·mL^−1^ for each enantiomer) in six replicates. For extraction recovery, the peak areas of each penconazole enantiomer and IS obtained from QC samples (spiked before extraction) were compared with those obtained from unextracted samples (spiked after extraction). In terms of matrix effect, the individual peak responses of each penconazole enantiomer and IS from unspiked samples (spiked after extraction) were compared with those prepared in standard solutions at the same concentrations. The IS-normalized matrix factor (NMF) was counted by using the equation as follow: NMF (%) = MF_analyte_/MF_IS_ × 100%.

The data of extraction recoveries and NMF were summarized in Table 3. For (+)-and (−)-penconazole, the extraction recoveries changed from 95.3% to 96.3% and 94.8% to 97.9% with all RSD values within 6.3%. The extraction recovery of IS was 90.9%, and the RSD was 5.4%. For (+)-and (−)-penconazole, the NMF was in the range of 103.6%–105.8% and 104.9%–109.1%, respectively, where the RSD was below 5.2%. Accordingly, the ion suppression or enhancement from the plasma matrix was negligible in this work.

#### 3.4.5. Stability

The stability of penconazole enantiomers in rat plasma was tested at three QC levels (5.0, 50.0, and 200.0 ng mL^−1^ for each enantiomer) in six replicates. Short-term stability (at room temperature for 12 h), postpreparative stability (in the autosampler at 4 °C for 24 h) and long-term stability (at −80 °C for 30 days) were investigated. Meanwhile, the freeze–thaw stability of plasma samples was also tested by three freezing and thawing cycles from −80 °C to room temperature. The results proved that penconazole enantiomers were stable during the stored time. It also demonstrated that no chiral inversion was observed between the two enantiomers during the experiment period.

### 3.5. Application to Enantioselective Toxicokinetic Study

The developed LC-MS/MS method was applied to the enantioselective toxicokinetic study of (+)-and (−)-penconazole after oral administration of 50.0 mg kg^−1^ rac-penconazole to female and male rats. The plasma concentration–time curves for (+)-and (−)-penconazole at different genders are shown in Figure 3. The relevant toxicokinetic parameters are summarized in Table 4. The toxicokinetic parameters of two penconazole enantiomers were compared by paired sample T-tests. Statistically significant difference existed between the parameters of (+)-and (−)-penconazole if *p* < 0.05.

It could be seen from Figure 3 that the concentration of (+)-penconazole enantiomer in rat plasma was always higher than that of its antipode in both male and female rats. According to the parameters listed in Table 4, there were significant differences in toxicokinetic parameters between two enantiomers, with a larger AUC and a smaller CL for (+)-penconazole than those for (−)-penconazole in the male. For female, the C_max_, AUC, MRT, t_1/2_ were significantly higher and the CL was lower for (+)-penconazole compared with those obtained for (−)-penconazole. Based on the above results, it can be speculated that the enantioselective toxicokinetics of penconazole enantiomers were more obvious in female rats compared with male rats. Additionally, the CL values of (+)-enantiomer in both sex rats were smaller than those of the (−)-enantiomer, indicating a lower metabolic of the (+)-enantiomer. These results indicated substantial stereoselectivity of the toxicokinetics of penconazole enantiomers in rats. The most likely reasons involved in these results may be enantioselective metabolism and excretion of penconazole. The distribution differences of two enantiomers in rat plasma may also contribute to enantioselective behavior of the chiral fungicide.

## 4. Conclusions

For the first time, a specific and sensitive LC-MS/MS method was developed for the determination of penconazole enantiomers in rat plasma and applied to the stereospecific toxicokinetic study. For the enantioseparation, a Chiralpak IC column was selected, and satisfactory resolution (R = 2.21) of two penconazole enantiomers was obtained under the reversed-phase mode. Critical validation parameters, including specificity, carry-over effect, linearity and the lower limit of quantification, accuracy and precision, extraction recovery, matrix effect, and stability, were all within the acceptable limits. After validation, the method was successfully applied to the enantioselective toxicokinetic study of penconazole enantiomers in rats. Following oral administration of 50 mg·kg^−1^ racemic penconazole to both sexes of SD rats, it was found that the AUC and clearance of (+)-penconazole were obviously higher than those of its antipode for male rats. For female rats, significant difference in the C_max_, t_1/2_, AUC, MRT, and clearance between the two enantiomers of penconazole was observed, which indicated the enantioselective toxicokinetic behavior of penconazole in vivo. Additionally, the enantioselective difference of penconazole enantiomers in female rats was more obvious compared that in male rats. Such data may provide evidence for the development and application of single-enantiomer penconazole in agriculture.

## Figures and Tables

**Figure 1 molecules-25-02964-f001:**
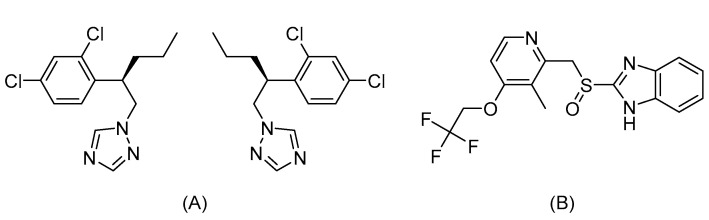
The chemical structure of penconazole (**A**) and l-lansoprazole (IS) (**B**).

**Figure 2 molecules-25-02964-f002:**
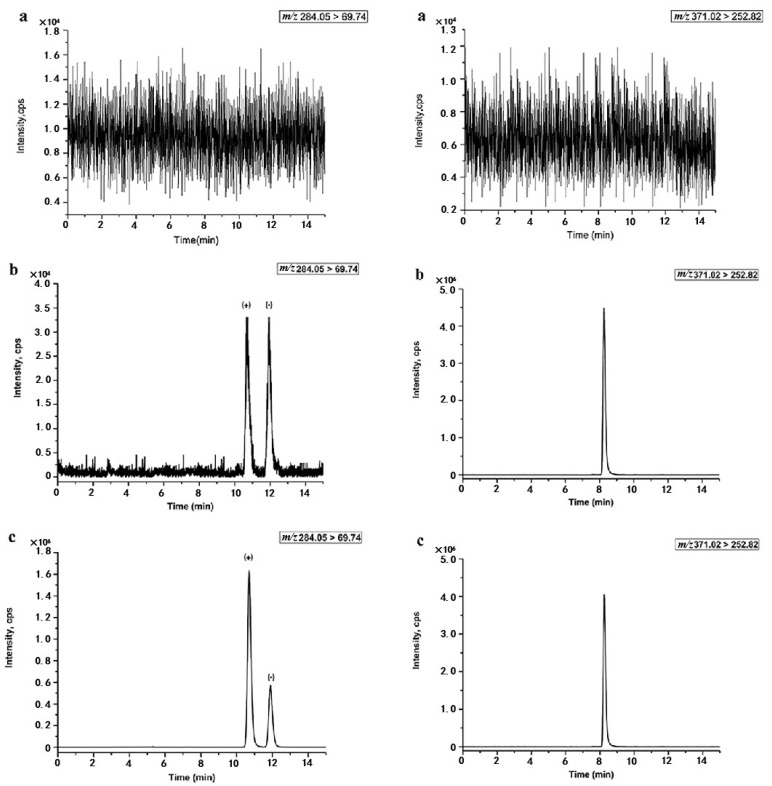
Typical chromatograms of (+)-and (−)-penconazole with L-lansoprazole (IS) in rat plasma sample. (**a**) Blank plasma sample; (**b**) blank plasma sample spiked with penconazole enantiomers at the lower limit of quantification (LLOQ) level; (**c**) extracted rat plasma sample at 2 h after single oral administration of 50 mg·kg^−1^ rac-penconazole to female rats.

**Figure 3 molecules-25-02964-f003:**
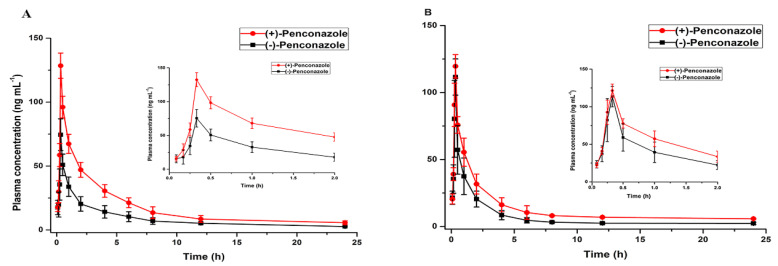
Plasma concentration–time curves obtained after a single oral administration of 50 mg·kg^−1^ penconazole to Sprague–Dawley rats: (**A**) Adult female; (**B**) adult male.

**Table 1 molecules-25-02964-t001:** The effect of mobile phase compositions on resolution (Rs) and retention time (t_R_).

Content of MeOH (%)	Retention Time (t_R_)	Resolution (Rs)	Content of CAN (%)	Retention Time (t_R_)	Resolution (Rs)
90	8.3	0.32	90	6.2	0.96
80	15.3	0.63	80	10.7	2.21
70	24.5	1.02	70	15.3	3.45
60	35.4	1.43	60	24.9	4.87

**Table 2 molecules-25-02964-t002:** Intraday and interday accuracy and precision of analysis method in spiked rat plasma at three quality control (QC) levels (*n* = 6).

Analytes	Spiked Concentration (ng mL^−1^)	Intraday			Interday		
		Measured Concentration (ng mL^−1^)	Precision (RSD, %)	Accuracy (RE, %)	Measured Concentration (ng mL^−1^)	Precision (RSD, %)	Accuracy (RE, %)
(+)-penconazole	5.0	5.1 ± 0.2	3.9	−0.7	5.0 ± 0.3	6.0	−1.1
	50.0	50.9 ± 2.4	4.7	1.4	51.0 ± 1.7	3.3	2.7
	200.0	196.9 ± 7.2	3.7	0.5	205.4 ± 2.5	1.2	0.9
(−)-penconazole	5.0	5.1 ± 0.2	3.9	1.4	5.2 ± 0.2	3.8	3.2
	50.0	50.1 ± 0.7	1.4	−0.9	50.9 ± 1.2	2.4	1.6
	200.0	201.6 ± 4.8	2.4	0.3	198.8 ± 6.0	3.0	−0.4

**Table 3 molecules-25-02964-t003:** The extraction recovery and matrix effect of penconazole enantiomers at three QC levels in rat plasma (*n* = 6).

Analytes	Concentration (ng mL^−1^)	Recovery (%, Mean ± SD)	RSD	NMF (%, Mean ± SD)	RSD
(+)-penconazole	5.0	95.3 ± 5.9	6.2	103.8 ± 4.1	3.9
	50.0	96.3 ± 3.5	3.6	105.8 ± 3.7	3.5
	200.0	95.8 ± 1.0	1.0	103.6 ± 3.4	3.3
(−)-penconazole	5.0	96.0 ± 6.0	6.3	109.1 ± 5.7	5.2
	50.0	97.9 ± 2.5	2.6	104.9 ± 4.6	4.4
	200.0	94.8 ± 2.4	2.5	107.1 ± 1.8	1.7
IS	100.0	90.9 ± 4.9	5.4	101.2 ± 2.8	2.8

**Table 4 molecules-25-02964-t004:** Toxicokinetic parameters (mean ± SD) of penconazole enantiomers in Sprague–Dawley rats (*n* = 6) following single oral administration at a dose of 50 mg kg^−1^.

Parameters	Male Rats			Female Rats		
	(+)-Penconazole	(−)-penconazole	(+)/(−)-penconazole	(+)-Penconazole	(−)-penconazole	(+)/(−)-penconazole
C_max_ (ng·mL^−1^)	119.4 ± 8.69	111.14 ± 13.51	1.07 ± 0.21	132.47 ± 10.37	75.59 ± 13.09	1.75 ± 0.46 a
T_max_ (h)	0.31 ± 0.04	0.30 ± 0.04	1.03 ± 0.23	0.32 ± 0.09	0.30 ± 0.11	1.07 ± 0.57
t_1/2_ (h)	8.89 ± 3.28	8.07 ± 4.61	1.10 ± 0.66	9.29 ± 3.76	3.36 ± 0.71	2.76 ± 0.59 a
AUC_0–24h_ (ng·h·mL^−1^)	263.12 ± 32	187.90 ± 43	1.40 ± 0.48 a	364.10 ± 79	175.64 ± 26	2.07 ± 0.19 a
AUC_0–∞_ (ng·h·mL^−1^)	294.30 ± 28	199.32 ± 50	1.48 ± 0.49 a	356.05 ± 89	185.54 ± 30	1.92 ± 0.19 a
MRT_0–24_ (h)	4.82 ± 0.25	4.31 ± 1.47	1.12 ± 0.45	5.07 ± 1.36	2.58 ± 0.80	1.97 ± 0.50 a
CLz/F (L/h/kg)	264,814 ± 16,328	779,535 ± 75,588	0.34 ± 0.00094 a	147,374 ± 37,892	970,193 ± 50,180	0.15 ± 0.08 a

Data are expressed as mean ± SD: (a) values significantly (*p* < 0.05) different from the concentration of (−)-penconazole; C_max_, maximum plasma concentration; T_max_, the time to reach the maximum plasma concentration; t_1/2_, half-life of elimination; AUC_0–24_, area under the plasma concentration curve from 0 to 24 h; AUC_0–∞_, area under the plasma concentration curve from 0 to infinity; MRT_0–24_, mean retention time from 0 to 24 h; CLz/F, apparent clearance.

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
