# Peer review of "Enantioseparation and Determination of Penconazole in Rat Plasma by Chiral LC-MS/MS: Application to a Stereoselective Toxicokinetic Study"

_molecules, 2020, doi:10.3390/molecules25132964_

Round 1

Reviewer 1 Report

Molecules_2020_849502

The manuscript untitled “Enantioseparation and determination of penconazole in rat plasma by chiral LC-MS/MS: application to a stereoselective toxicokinetic study” by Siman Ma, Jia Lun, Yanru Liu, Zhen Jiang, and Xingjie Gu.

The manuscript represents a research article and is devoted to enantioseparation and evaluation of penconazole enantiomers in rat plasma.

Five-membered nitrogen heterocycles, the so-called azoles including imidazole, pyrazole, triazole, adenine (imidazole cycle as part of purine) are the central ingredients in many drugs and others bioactive compounds. Indeed, 1,2,4-triazole as a part of Penconazole represents a chiral molecule, widely used as a fungicide.

The authors prepared a very serious and thorough study, which seems important for evaluation of bio-effects of both penconazole enantiomers. Toxicological studies of enantiomers are a trend in modern bio-research.

These materials could be useful for biological chemistry and medicinal chemistry.

The manuscript does not raise any objections, and can be published in Molecules after minor revision.

Several details and inaccuracies should be noted.

  1. Abstract. Why is the correlation coefficient defined as R2?
  2. Introduction. It is quite appropriate to give the formulas of pencoazole and its two enantiomers in the "Introduction" section.
  3. Is penconazole in racemic form available in industry for agriculture? It should be clarified.
  4. Line 55. Firs author of citing reference [10] is Meng, but not Zhang.
  5. Experimental. Line 95. What is the origin of the (-)-enantiomer of penconazole?
  6. Line 96. …from 25.0 to 2500.0 Really?
  7. Results and discussion. Since the article is supposed to be published in the chemical journal Molecules, I would like to recommend some explanations regarding the parameters studied in terms of toxicokinetic studies, as well as comparisons with the literature data for other compounds studied (Section 3.2).
  8. Conclusion. Should be corrected. The first paragraph repeats twice “For the first time”.
  9. In the Abstract authors made a comment “This is the first experiment regarding the stereospecific toxicokinetic study of penconazole”. In Conclusion it is necessary to comment on this statement. Kinetics indicates a process in time. What happens to both enantiomers of penconazole in the study period?
  10. References. References and their citation should be verified.
  11. For reference [10] should be given year (2019) of publication.

Reviewer 2 Report

The study presents an analytical method for the separation of the 2 enantiomers of penconazole in rat plasma by chiral LC-MS/MS. The application in a toxicokinetic study adds value to the study.

Anyway the study description should be impounded. Please consider the following:

R37-38. Penconzole, next to other pesticides, fungicides, pharmaceuticals, personal care production, etc, represent what we call at the present ‘ emerging contaminates’ or ‘ pollutants of emerging concern’. ‘Inevitable trend’ sound a little bet inappropriate. Please rephrase.

R 42. ‘pesticide accumulation for a long time…’- reference needed. The phrase is too general. Give an example of something similar related to triazoles, the class to which penconazole belongs. For example it is known that triazole affect aromatase system (CYP19). E. R. Trösken, K. Scholz, R. W. Lutz, W. Völkel, J. A. Zarn, W. K. Lutz, Comparative Assessment of the Inhibition of Recombinant Human CYP19 (Aromatase) by Azoles Used in Agriculture and as Drugs for Humans, Endocrine Research,  30, No. 3, (2004), 387-394.

R48. ‘…….chiral pesticides may differ in biological activities and toxicities’ Reference needed.

R 55. Capital letter for Zhang et al.

R 76. It is not clear and sounds like a contradiction with R75.  You affirm that ‘The Chiralpak IC column was selected to separate 78 penconazole enantiomers in reversed-phase mode for the first time’.

Few rows above ‘penconazole enantiomers achieved basic separation under reversed-phase conditions.’ By Wang et al., 2014 and ‘Zhao et al.[12] separated 70 penconazole enantiomers in reversed-phase mode using an amylose based CSP (Chiralpak IG)’

It is not clear what are the differences between your study and the mentioned references. I understand that you used a cellulose based column (IC) in reverse phase chromatography, while other used the same column under normal-phase or amylose based (IG) under reverse phase.

I think you should rephrase to highlight the novelty of your study and emphasis the differences between columns.

R120. How you confirmed the identity of the enantiomers?

R 158. ‘three polysaccharide-based columns were tested to separate penconazole enantiomers’ you should include the test results and analysis of the parameters  on which the choice of column was based

R185. Optimization of sample pretreatment procedures

‘pretreatment methods including solid-phase extraction (SPE), liquid-liquid extraction (LLE) and protein precipitation (PPT) were investigated in this study’.

The result of those tests and the comparative analysis of the results should be included at least in the supplementary files.

Round 2

Reviewer 2 Report

The manuscript was improved and, in my opinion, can be published in the present form.